# The Flow Stress–Strain and Dynamic Recrystallization Kinetics Behavior of High-Grade Pipeline Steels

**DOI:** 10.3390/ma15207356

**Published:** 2022-10-20

**Authors:** Lei Wang, Lingkang Ji, Kun Yang, Xiongxiong Gao, Hongyuan Chen, Qiang Chi

**Affiliations:** State Key Laboratory for Performance and Structure Safety of Petroleum Tubular Goods and Equipment Materials, CNPC Tubular Goods Research Institute, Xi’an 710077, China

**Keywords:** the flow stress, dynamic recovery, dynamic recrystallization, kinetics, high-grade pipeline steel

## Abstract

The hot deformation behavior of high-grade pipeline steels was studied in the strain rate range of 0.001~0.1 s^−1^ and the temperature range of 1050~1200 °C by using hot compression tests on a Gleeble 3500 thermomechanical simulator. The flow stress increases with the increase in strain rate and the decrease in deformation temperature, and the deformation activation energy is about 358 kJ/mol. The flows stress–strain behavior of the work-hardening and dynamic recovery (DRV) was calculated using the Estrin–Mecking equation, and the kinetics model of the dynamic recrystallization (DRX) was established based on the Avrami equation through characteristic strains. Furthermore, the flow stress–strain behavior of high-grade pipeline steels was predicted by the established model based on the coupling effects of DRV and DRX. The corresponding predicted results are in good agreement with the experimental results according to standard statistical parameters analysis. Finally, the economic strain (*ε*_3_) is proposed by the third derivative of the given kinetic model. Based on these calculation results, when the economic strain (*ε*_3_) is reached, uniform and refined DRX grains can be obtained, the energy consumption reduced, and the production costs controlled, which is of great significance to actual factory production.

## 1. Introduction

Due to the combination of high strength and toughness, high-grade pipeline steels, such as, X70, X80, and X100, has been extensively studied [1,2,3,4,5]. Thermo-mechanical control process (TMCP) is an indispensable and important technology in the manufacture of high-grade pipeline steels, which mainly involves the thermal deformation process of steel plates. During the thermal deformation process, the flow stress reflects the difficulty of deformation of the material, and also puts forward corresponding requirements for the equipment capability. Thus, the accurate description of the flow behavior under different deformation conditions is of great significance for the design of the pipeline steel manufacturing process (excellent product performance) and the selection of manufacturing equipment (less energy consumption). Usually, a constitutive equation is used to describe the flow behavior of materials, such as the Arrhenius equation being applied to nickel-based alloys [6,7,8], steels [9,10,11], Ti alloys [12,13], and Al alloys [14,15]. The Johnson–Cook equation is used for a wide range of strain rates (e.g., 10^−^^4^ up to 10^4^ s^−^^1^ [16,17,18]). The above method is called the phenomenological constitutive equation based on mathematical function fitting methods, and lacks obvious physical meaning [12,18]. On the other hand, under the evaluation of microstructure and deformation mechanisms of the materials, the flow stress was predicted by the Kocks–Estrin equation through considering the evolution of dislocation density [19,20]. However, the Kocks–Estrin equation can only describe the work-hardening and dynamic recovery (DRV). Under high temperature, the flow stress of many metals tends to have obvious stress peaks, meaning that the deformation mechanism is dominated by dynamic recrystallization, rather than a DRV. Generally, DRX consists of the process of nucleation and the grain boundary mobility, which can be described using the Avrami equation [21]. For example, the DRX kinetic behavior of a low-carbon steel during the hot deformation was studied using the Avrami model [22], where the volume fraction of DRX was predicted accurately. Based on the Avrami equation, the effect of the hot deformation parameters on the DRX behavior of X70 pipeline steel was studied and the evolution of volume fraction of DRX was described as a function of the different temperatures and strain rates [5]. A DRX kinetic model of the low-alloyed and micro-alloyed steels was proposed by Hernandez [23,24], and the flow stress–strain curve can also be predicted through the improved Avrami equation. This shows that the DRX behavior of pipeline steel can be accurately described using the Avrami equation. However, the establishment of a DRX kinetic model is only to predict the volume fraction of DRX. The evolution of the transform velocity of DRX is not analyzed in depth, and more useful information for the preparation process of pipeline steel cannot be obtained.

In this paper, X70 pipeline steel is the subject of research and the constitutive relationship between the flow stress and deformation parameters is established. The characteristic strains/stresses are obtained by the work-hardening rate curve. The volume fraction of DRX is obtained by the difference between the constructed flow stress curves of DRV and the experimental stress–strain curve. Then, the kinetic model is established based on the Avrami equation. The flow stress of X70 pipeline steel under different deformation conditions can be predicted by considering the coupling between the work-hardening behavior and the dynamic softening behavior of DRV and DRX. Finally, the economic strain (*ε*_3_) is proposed by the third derivative of the given kinetic model of DRX for guaranteeing uniform and fine grains, saving energy consumption, and reducing production cost.

## 2. Materials and Methods

The chemical composition (wt%) of the X70 pipeline steel used in this work was as follows: C 0.05, Si 0.16, Mn 1.75, P 0.008, S 0.0024, Ni 0.1, Nb 0.053, V 0.0051, Cu 0.03, Ti 0.015. Before isothermal constant strain rate compression tests carried out on a Gleeble 3500 thermomechanical simulator were performed, all the specimens were heated to 1200 °C, held for 1 h, and then quenched to room temperature. The compression specimens were machined to a diameter of 10 mm and a length of 15 mm. To minimize the friction during hot deformation, tantalum foils were applied between the compression specimens and the platens. The compression specimens were heated to 1200 °C at a rate of 30 °C/s, holding for 300 s, then cooled to different deformation temperatures (1050~1150 °C) at a rate of 10 °C/s. All specimens were compressed to a true strain 0.7, then immediately water-cooled down to room temperature. The thermomechanical processing schedule is schematically shown in Figure 1.

## 3. Results

### 3.1. Flow Stress Curves

The flow stress–strain curves of the X70 pipeline steel at different deformation temperatures and strain rates in the isothermal compression tests are shown in Figure 2. During the hot deformation process, the shape of the flow stress curves is the result of competition between work-hardening and dynamic softening (DRV and DRX). As shown in Figure 2, at the initial stage of deformation, the flow stress increases rapidly due to the strong work-hardening behavior, based on the dislocation density, increases rapidly with small strain. When the critical strain is reached, the flow stress increases very slowly due to the occurrence of DRX softening, in which the nucleation process of DRX can consume a large number of dislocations. As the strain continue to increase, the flow stress peak occurs when the work-hardening is equal to the DRX softening. Thereafter, dynamic softening is greater than the work-hardening, and the flow stress gradually decreases until steady-state stress is reached. It can be found that the flow stress curve drops with the increase in temperature and with the decrease in the strain rate [25]. It is mainly because hot deformation is a thermally activated process. High temperature and low strain rate can promote the thermally activated process, reduce the critical stress for crystal grain slip, and enhance the recrystallized grain boundary migration, resulting in decreased work-hardening, as well as increased dynamic softening (DRV, DRX) [8,9,10,11].

### 3.2. The Characteristic Stress/Strain

Usually, the characteristic stress of DRX softening curve can be obtained based on the working hardening rate (θ = dσ/dε) curve, including the critical stress (σ_c_), the peak stress (σ_p_), the maximum softening stress (σ_m_), and the steady state stress (σ_ss_) [22]. A typical total θ-σ curve of X70 pipeline steel is shown in Figure 3a, where, σ_p_, σ_m,_ and σ_ss_ can be easily obtained. In order to determine the critical stress (σ_c_) for the onset of DRX, Poliak and Jonas [26] proposed the double-differentiation method. Firstly, the local flow stress–strain curves from the yield stress (σ_0_) to the peak stress (σ_p_) were fitted using the seventh order polynomial. The working hardening rate curves under different deformation conditions are shown in Figure 3b. Then, when the derivative of the work-hardening rate curve is equal to 0, the critical stress (σ_c_) is obtained under different deformation conditions, as shown in Figure 3c. All characteristic strains of DRX can be obtained from corresponding characteristic stresses in the flow stress curve.

Since the peak stress/strain can be directly and accurately obtained from the experimental flow stress curve, the empirical relationship between the peak stress/strain and other characteristic stress/strain is often established. It can be seen from Figure 4a that the yield strain/stress, the critical stress/strain, the maximum softening stress/strain, and the steady state stress/strain show a good linear relationship with the peak strain, in which their empirical relationship can simply expressed as: ε_c_ = 0.51ε_p_, ε_max_ = 2.52ε_p_, and ε_ss_ = 3.93ε_p_. Similarly, characteristic stresses are in a certain proportion to the peak stress (Figure 4b), which can be expressed as: σ_c_ = 0.89σ_p_, σ_0_ = 0.61σ_p_, σ_ss_ = 0.78σ_p_, and σ_sat_ = 1.17σ_p,_ respectively.

To establish the relationship between characteristic stresses/strains and processing parameters (deformation temperatures and strain rates), the constitutive equation can be used and expressed as follows:(1)$$\dot{\varepsilon }=A_{1}\sigma_{p}^{m^{*}}\exp\left(-\frac{Q_{d}}{RT}\right)$$
(2)$$\dot{\varepsilon }=A_{2}\exp\left(\beta \sigma_{p}\right)\exp\left(-\frac{Q_{d}}{RT}\right)$$
(3)$$\dot{\varepsilon }=A{\left[\sinh\left(\alpha \sigma_{p}\right)\right]}^{m}\exp\left(-\frac{Q_{d}}{RT}\right)$$

Usually, the relationship between the deformation temperature and strain rate can be established through Zener–Hollomon parameter (*Z*) as follows [27]:(4)$$Z=\dot{\varepsilon }\exp\left(\frac{Q_{d}}{RT}\right)=A{\left[\sinh\left(\alpha \sigma_{p}\right)\right]}^{m}$$

In Equations (1)–(4), *Q_d_* is the deformation activation energy, *R* is the gas constant (8.314 Jmol*^−^*^1^K*^−^*^1^), and *A*, *A*_1_, *A*_2_, *β*, *α*, *m*, *m** are the material constants. Firstly, the value of *α* in Equation (3) should be calculated as 0.0124 by using the relationship between *β* and *m** (*β*/*m**), where the average value of *β* and *m** can be calculated as 0.07 and 5.39 by linear fitting, respectively. Then, the deformation activation energy (*Q_d_*) can be obtained by partial differentiation of Equation (3):(5)$$Q_{d}=R{\left[\frac{\partial \dot{\ln\varepsilon }}{\partial \ln\left(\sinh\left(\alpha \sigma_{p}\right)\right)}\right]}_{T}{\left[\frac{\partial \ln\left(\sinh\left(\alpha \sigma_{p}\right)\right)}{\partial \left(1/T\right)}\right]}_{\dot{\varepsilon }}=RmS$$

The values of m can be obtained as 4.73 by calculating the average value of slopes under different deformation temperatures in Figure 5a, and the value of *Q_d_* can be derived as 358 KJ/mol by calculating the average value of slops under different strain rates in Figure 5b. Finally, the constitutive equation of the X70 pipeline steel can be expressed as following:(6)$$\dot{\varepsilon }=1.8123\times 10^{16}{\left[\sinh0.0124\right]}^{4.73}\exp\left(-\frac{358,656}{RT}\right)$$

Furthermore, the peak stress can be calculated by using Equation (7).
(7)$$\sigma_{p}=123.153\times \ln\left\{{\left[\frac{Z}{1.8123\times 10^{16}}\right]}^{1/4.73}+{\left[{\left(\frac{Z}{1.8123\times 10^{16}}\right)}^{2/4.73}+1\right]}^{1/2}\right\}$$

**Figure 5 materials-15-07356-f005:**
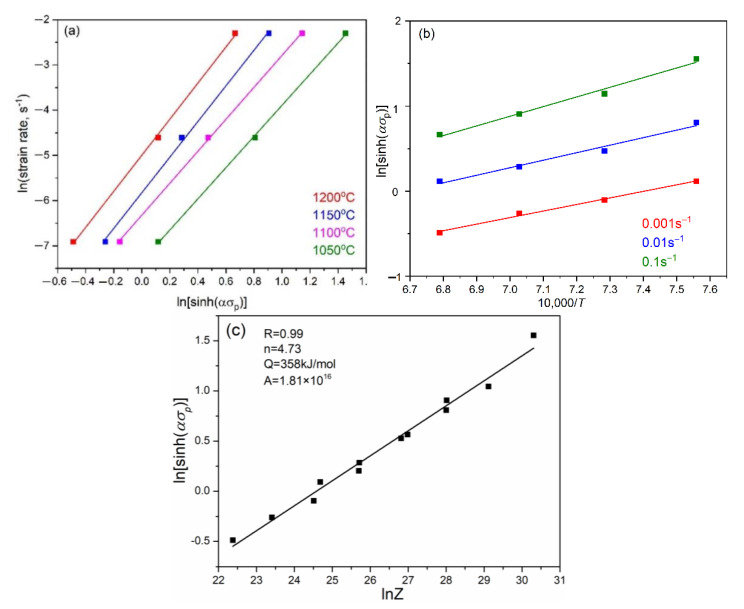
Relationship of ln[sinh(*ασ*_p_)] vs. ln$\dot{\varepsilon }$ (**a**); ln[sinh(*ασ*_p_)] vs. 1/*T* (**b**) and the peak stress vs. *Z* (**c**) deduced from the X70 pipeline steel subjected to different deformation conditions.

Furthermore, based on the exponential relationship in Equation (7), a relatively simple exponential relationship between characteristic stresses/strains and *Z* parameter can be directly obtained, which is expressed as follows:(8)$$\sigma_{p}=aZ^{b}\;\;\;\;\;\;\;\;\;\;\sigma_{x}=A\sigma_{p}=A*aZ^{b}$$
(9)$$\varepsilon_{p}=cZ^{d}\;\;\;\;\;\;\;\;\;\;\varepsilon_{x}=C\varepsilon_{p}=C*cZ^{d}$$
where *a*, *b*, *c* and *d* are the material constants. The function relation between the peak stress/strain (σ_p_/ε_p_) and *Z* parameter can be obtained through linear regression. Furthermore, the relationship between other characteristic stresses and *Z* can also be determined one by one by using the above proportional relation (Equations (8) and (9)). The calculated results of the function relation are in Table 1. 

### 3.3. The Flow Behavior in DRV

The flow stress curve of DRV regime during hot deformation can be described by the competition between storage and annihilation process of dislocation. It can be expressed as follows [22,28]:(10)$$\frac{d\rho }{d\varepsilon }=h-\mathrm{rp}$$
where the first term on the right-hand side of Equation (10) represents the contribution of the work-hardening and the second term due to DRV. The work-hardening term can be regarded as constant (h) with respect to the strain; meanwhile, a parameter (r) of DRV follows the first order kinetics. When the initial dislocation density is denoted by ρ_0_, the evolution of dislocation density in the wok-hardening and DRV regime can be described by integrating Equation (10):(11)$$\rho =\rho_{0}\exp\left(-r\left(\varepsilon -\varepsilon_{o}\right)\right)+\frac{h}{r}\left(1-\exp\left(-r\left(\varepsilon -\varepsilon_{0}\right)\right)\right)$$

The relationship between the flow stress and the dislocation density is usually expressed as:(12)$$\sigma =\alpha \mu b\sqrt{\rho }$$
where α is a material constant, μ is the shear modulus, and b is the magnitude of Burger’s vector. The value of the flow stress of DRV can be calculated by combining Equations (11) and (12), and is expressed as follows:(13)$$\sigma_{\mathrm{rec}}={\left(\sigma_{0}^{2}\exp\left(-r\left(\varepsilon -\varepsilon_{0}\right)\right)+{\left(\alpha \mu b\right)}^{2}\left(\frac{h}{r}\right)\left(1-\exp\left(-r\left(\varepsilon -\varepsilon_{0}\right)\right)\right)\right)}^{\frac{1}{2}}$$
where σ_rec_ represents the flow stress of DRV, σ_0_ the yield stress, and ε_0_ the strain corresponding to σ_0_. The relationship between the stress and the strain under DRV duration can be described as Equation (13). With the increase in the strain (ε), the stress (σ_rec_) tends toward a saturation value (σ_sat_) corresponding to an equilibrium between dislocation storage and dislocation annihilation, i.e., dρ/dε = 0, $\sigma_{\mathrm{rec}}=\alpha \mu b\sqrt{h/r}$. After the saturated stress (σ_sat_) is brought into Equation (13), it can be expressed as the following formula:(14)$$\sigma_{\mathrm{rec}}={\left(\sigma_{\mathrm{sat}}^{2}-\left(\sigma_{\mathrm{sat}}^{2}-\sigma_{0}^{2}\right)\exp\left(-r\left(\varepsilon -\varepsilon_{0}\right)\right)\right)}^{\frac{1}{2}}$$

All the stress/strain parameters in Equation (14) were obtained in Section 3.2, so, as long as the r parameter is determined, the flow stress–strain curve of DRV can be obtained. The parameter r can be determined based on the differentiation of Equation (14) with respect to ε, and multiplication by σ_rec_ leads to the following relation [29]:(15)$$\frac{{d\sigma }_{\mathrm{rec}}}{d\varepsilon }\sigma_{\mathrm{rec}}={\theta \sigma }_{\mathrm{rec}}=\frac{1}{2}{r\sigma }_{\mathrm{sat}}^{2}-\frac{1}{2}{r\sigma }_{\mathrm{rec}}^{2}$$
where θ is the work-hardening rate prior to the critical strain (ε_c_). In order to obtain the value of parameter r, the calculation formula is proposed by deriving Equation (16) with respect to $\sigma_{\mathrm{rec}}^{2}$:(16)$$K=\frac{d\left({\theta \sigma }_{\mathrm{rec}}\right)}{{d\sigma }_{\mathrm{rec}}^{2}}=-\frac{1}{2}r$$

According to Equation (16), the value of parameter r under different deformation conditions can be calculated as −2K and is shown in Figure 6, in which solid line is calculated from the experimental flow stress curve prior to σ_c_. Meanwhile, the saturation stress (σ_sat_) can also be obtained by the intersection of the dashed line and the abscissa indicated by arrows in Figure 6a–c.

A functional relationship between the parameter *r* and different deformation conditions (*Z*) is established and is shown in Figure 6d. The parameter *r* decreases with the increase in the *Z* parameter, indicating that the dynamic recovery process is more difficult with the decrease in deformation temperature or the increase in strain rate. This is consistent with the research results of Jonas et al. [22], indicating that the occurrence difficulty of dynamic recovery process increases with the increase in stress value.

Based on the above analysis, the flow stress–strain curve of DRV for X70 pipeline steels can be summarized as:(17)$$\left\{\begin{array}{c} \varepsilon_{0}=0.00171Z^{0.118}\\ \sigma_{0}=0.606Z^{0.218}\\ \sigma_{sat}=1.162Z^{0.218}\\ r=186.233Z^{-0.124}\\ \sigma_{\mathrm{rec}}={\left(\sigma_{sat}^{2}-\left(\sigma_{sat}^{2}-\sigma_{0}^{2}\right)\exp\left(-r\left(\varepsilon -\varepsilon_{0}\right)\right)\right)}^{\frac{1}{2}} \end{array}\right.$$

Figure 2a–c are the prediction results, and it can be seen that the calculated flow stress curve of DRV at all deformation conditions coincide well with the experimental flow stress–strain curves before the critical stress/strain. After the critical stress/strain, the flow stress of DRV keeps increasing until it reaches the saturation stress (σ_sat_). On the whole, the flow stress–strain curve of DRV increases with the decrease in temperature or the increase in strain rate.

### 3.4. Kinetic Model of DRX

Figure 7 is a schematic diagram of the dynamic softening of flow stress at elevated temperature. After the critical stress, the flow stress presents two different characteristics represented by the “blue” and “read” lines in Figure 7, which are attributed to DRV and DRX, respectively. For the “blue” curve, when the work-hardening rate is equal to the DRV softening rate, the stress reaches saturation value and remains constant (σ_sat_). For the “red” curve, after the flow stress reaches the peak stress, the DRX softening rate is stronger than the work-hardening rate resulting in the flow decreasing continuously to the steady state value (σ_ss_). Therefore, using the difference between the experimental stress curve and the flow stress of DRV, the volume fraction of the DRX can be expressed as:(18)$$X_{DRX}=\frac{\Delta \sigma }{\sigma_{\mathrm{sat}}-\sigma_{\mathrm{ss}}}=\frac{\sigma_{\mathrm{rec}}-\sigma_{\mathrm{drx}}}{\sigma_{\mathrm{sat}}-\sigma_{\mathrm{ss}}}$$

According to Equation (19), the volume fraction of DRX at the different deformation conditions can be calculated easily, and this method is particularly useful for high-grade pipeline steels, where metallographic microstructures are difficult to carry out. As shown in Figure 8, with the increase in strain, the volume fraction of DRX firstly increases and then decreases, which shows a typical S-shaped curve.

When the critical strain (ε_c_) is reached, the dislocation density near grain boundary is sufficient to promote the nucleation of DRX grains. With the increase in strain, DRX nucleation start to grow through the migration of new grain boundary. When the DRX transition is fully completed, the flow stress does not change with the increase in the strain, that is, the steady state stress occurs at this time. In this study, the volume fraction of DRX can be expressed generally as [29,30]:(19)$$X_{\mathrm{DRX}}=1-\exp\left(-0.693{\left(\frac{\varepsilon -\varepsilon_{c}}{\varepsilon_{0.5}-\varepsilon_{c}}\right)}^{n}\right)$$

Taking logarithm of Equation (19) gives:(20)$$\ln\left(-\ln\left(1-X_{\mathrm{DRX}}\right)\right)=\ln0.693+\mathrm{nln}\left(\frac{\varepsilon -\varepsilon_{c}}{\varepsilon_{0.5}-\varepsilon_{c}}\right)$$

According to Equation (20), the average calculation result of n at all deformation conditions is 1.88, as shown in Figure 9, which is in consistent with the range of 1.3~2.5 observed by other works [22,24,29,30].

Based on the above analysis, the kinetics equation of DRX for the X70 pipeline steel can be summarized as:(21)$$\left\{\begin{array}{c} \varepsilon_{c}=0.000434Z^{0.118}\\ \varepsilon_{0.5}=0.181Z^{0.118}\\ X_{\mathrm{DRX}}=1-\exp\left(-0.693{\left(\frac{\varepsilon -\varepsilon_{c}}{\varepsilon_{0.5}-\varepsilon_{c}}\right)}^{1.88}\right) \end{array}\right.$$

The comparison of the calculated and experimental result is shown in Figure 9. The calculated results are consistent with the experimental ones, indicating that the established kinetic equation can well-describe the DRX behavior of X70 pipeline steel during hot deformation process.

## 4. Application and Discussion

### 4.1. Prediction of Flow Stress Curves in DRV

The experimental flow stress curve can be predicted through incorporation between the DRV curve and DRX softening. By substituting Equations (19) and (21) into Equation (18), the flow stress–strain curve of high-grade pipeline steel can be obtained as follows:(22)$$\sigma ={\left(\sigma_{\mathrm{sat}}^{2}-\left(\sigma_{\mathrm{sat}}^{2}-\sigma_{0}^{2}\right)\exp\left(-r\left(\varepsilon -\varepsilon_{0}\right)\right)\right)}^{\frac{1}{2}}-\left(\sigma_{\mathrm{sat}}-\sigma_{\mathrm{ss}}\right)\left(1-\exp\left(-0.693{\left(\frac{\varepsilon -\varepsilon_{c}}{\varepsilon_{0.5}-\varepsilon_{c}}\right)}^{1.88}\right)\right)$$

Comparison between the calculation and experimental results are shown in Figure 10, in which the scatter points and solid line represent the results of calculations and experiment results, respectively. It can be found that the calculated flow stress curves from Equation (22) are in good agreement with the experiment the flow curves. In order to quantitatively describe the prediction accuracy of the model, the correlation coefficient (*R*) and average absolute relative error (*AARE*) are used, and expressed as:(23)$$R=\frac{\sum_{i=1}^{N}\left(E_{i}-\bar{E}\right)\left(C_{i}-\bar{C}\right)}{\sqrt{\sum_{i=1}^{N}{\left(E_{i}-\bar{E}\right)}^{2}\sum_{i=1}^{N}{\left(C_{i}-\bar{C}\right)}^{2}}}$$
(24)$$AARE\left(\%\right)=\frac{1}{N}\overset{N}{\underset{i=1}{\sum }}\left|\frac{E_{i}-C_{i}}{E_{i}}\right|\times 100\%$$
where *E_i_* and *C_i_* represents the experimental data and the calculated results, respectively. *E* and *C* are the mean values of *E_i_* and *C_i_*, respectively. *N* is the total number of data. The calculated result is shown as in Figure 10d, which shows a good correlation between the experimental and the calculated data through using Equation (22). The value of *R* and *AARE* for prediction equation are 0.999 and 1.95%, respectively. These results indicate that Equation (22) can predict accurately the flow stress-strain curve of the X70 pipeline steel.

### 4.2. Determination of Economic Strain (ε_3_)

Through Equation (20), the velocity equation of DRX can be expressed by Equation (25) and, furthermore, the quadratic derivative of Equation (25) was constructed and when it is 0:(25)$$\frac{dX_{DRX}}{d\varepsilon }=\frac{0.693n}{\left(\varepsilon_{0.5}-\varepsilon_{c}\right)}(1-X_{DRX}){\left(\frac{\varepsilon -\varepsilon_{c}}{\varepsilon_{0.5}-\varepsilon_{c}}\right)}^{n-1}$$
(26)$$\frac{d^{3}X_{DRX}}{d\varepsilon^{3}}=\frac{0.693n}{{\left(\varepsilon_{0.5}-\varepsilon_{c}\right)}^{3}}(1-X_{DRX}){\left(\frac{\varepsilon -\varepsilon_{c}}{\varepsilon_{0.5}-\varepsilon_{c}}\right)}^{n-3}\left\{\begin{array}{l} n^{2}{\left[0.693{\left(\frac{\varepsilon -\varepsilon_{c}}{\varepsilon_{0.5}-\varepsilon_{c}}\right)}^{n}\right]}^{2}\\ -3n(n-1)\left[0.693{\left(\frac{\varepsilon -\varepsilon_{c}}{\varepsilon_{0.5}-\varepsilon_{c}}\right)}^{n}\right]\\ -(n-2)(n-1) \end{array}\right\}=0$$

The solution of Equation (26) is
(27)$$0.693{\left(\frac{\varepsilon -\varepsilon_{c}}{\varepsilon_{0.5}-\varepsilon_{c}}\right)}^{n}=\frac{3n(n-1)\pm n\sqrt{\left(5n-1)(n-1\right)}}{2n}$$

The final strain is expressed as the following equation:(28)$$\varepsilon =\sqrt[{n}]{\frac{3n(n-1)\pm n\sqrt{\left(5n-1)(n-1\right)}}{1.386n}}(\varepsilon_{0.5}-\varepsilon_{c})+\varepsilon_{c}$$

The strain in Equation (28) corresponds to two inflection points of the transform velocity of DRX, the range of which includes the strain corresponding to the maximum velocity of DRX. Thus, there are three key strains, which are rewritten as:(29)$$\left\{\begin{array}{c} \begin{array}{l} \varepsilon_{1}=\sqrt[{n}]{\frac{3n(n-1)-n\sqrt{\left(5n-1)(n-1\right)}}{1.386n}}(\varepsilon_{0.5}-\varepsilon_{c})+\varepsilon_{c} \end{array}\\ \varepsilon_{2}=\varepsilon_{\begin{array}{l} \max \end{array}}\\ \varepsilon_{3}=\sqrt[{n}]{\frac{3n(n-1)+n\sqrt{\left(5n-1)(n-1\right)}}{1.386n}}(\varepsilon_{0.5}-\varepsilon_{c})+\varepsilon_{c} \end{array}\right.$$

The typical transform velocity curve of DRX for X70 pipeline steel is shown in Figure 11, in which it can be divided into three stages based on two inflection points (Equation (29)). In stage I, the DRX process is at the beginning, and the volume fraction and transform velocity of DRX are low, meaning that the developing process of DRX is just beginning. In order for the material to fully undergo DRX behavior, the strain should continue to increase and enter the second stage of recrystallization velocity curve. In stage II, the transition velocity of DRX increases quickly until it reaches the maximum, and then it decreases quickly, which indicates that the maximum transform rate of DRX is located at the second stage. When the strain is around ε_m_, the fraction volume of DRX is about 50%, and the mean grain size is rapid and shows uniform refinement. Until the strain increases to ε_3_, the volume fraction of DRX is greater than 90%, and the transformation process of DRX is basically completed. When the strain is greater than ε_3_, the developing process of DRX has slow velocity, which means that the complete transformation of the remaining approximately 10% of the DRX grains in stage III requires a huge amount of energy and time compared to stage II, resulting in a waste of resources and a decrease in efficiency. In order to obtain optical grain size, enhance production efficiency, and reduce manufacturing cost, ε_3_ is the very important economic strain that can guarantee finer and more uniform DRX grains, lower the energy consumption, and provide high production efficiency.

## 5. Conclusions

In this study, the flow stress–strain curves and DRX kinetic behavior of X70 pipeline steel were investigated using the hot compression tests in a temperature range of 1050~1200 °C and a strain rate range of 0.001~0.1 s^−1^. The flow stress increases with decreasing temperature and increasing strain rate, and the deformation activation energy is calculated as 358 kJ/mol. Using the relationship between characteristic strain and *Z* parameter, volume fraction of DRX was established according to the Avrami equation. Furthermore, the flow stress curve predicted by considering the coupling effect of the DRV and DRX processes is in good agreement with experimental results of X70 pipeline steel under different deformation conditions. Finally, through a proposed kinetics model, the most appropriate and economic strain (ε_3_) is obtained, which can guarantee fine and uniform equiaxed grain, high production efficiency, and low energy consumption.

## Figures and Tables

**Figure 1 materials-15-07356-f001:**
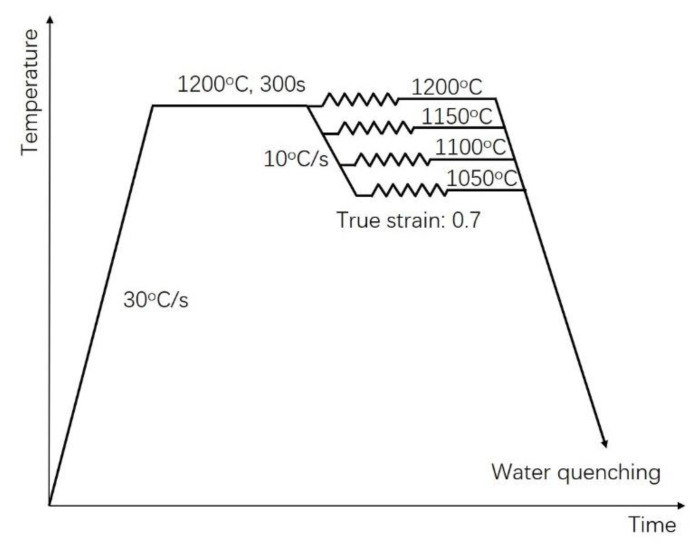
Schematic diagram of thermomechanical processing schedule.

**Figure 2 materials-15-07356-f002:**
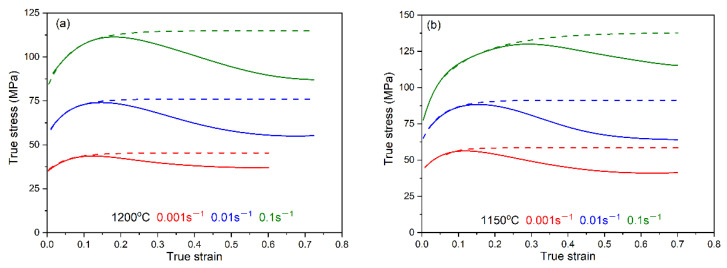
Experimental flow stress–strain curves of the X70 pipeline steel under different strain rates of (**a**) 1200 °C, (**b**) 1150 °C, (**c**) 1100 °C, (**d**) 1050 °C.

**Figure 3 materials-15-07356-f003:**
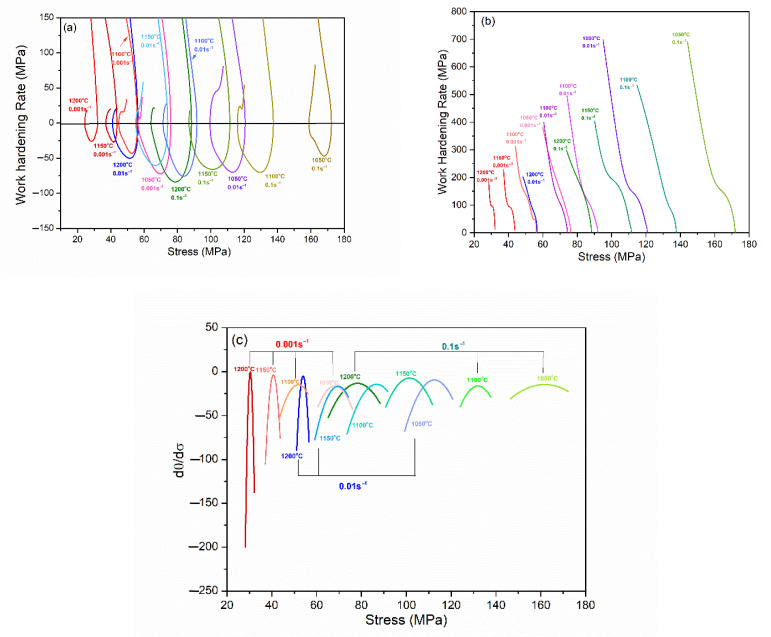
The relationship between work-hardening rate and true stress of X70 pipeline steel in different deformation conditions: (**a**) whole curve; (**b**) local curve from yield to peak stress stage; (**c**) the second derivative of the work-hardening rate curve.

**Figure 4 materials-15-07356-f004:**
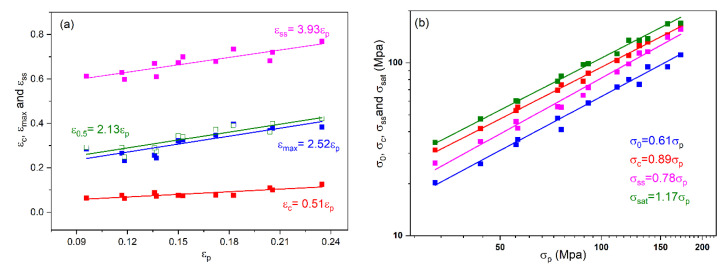
Relationship between characteristic parameter (strain (**a**) and stress (**b**)) and Z.

**Figure 6 materials-15-07356-f006:**
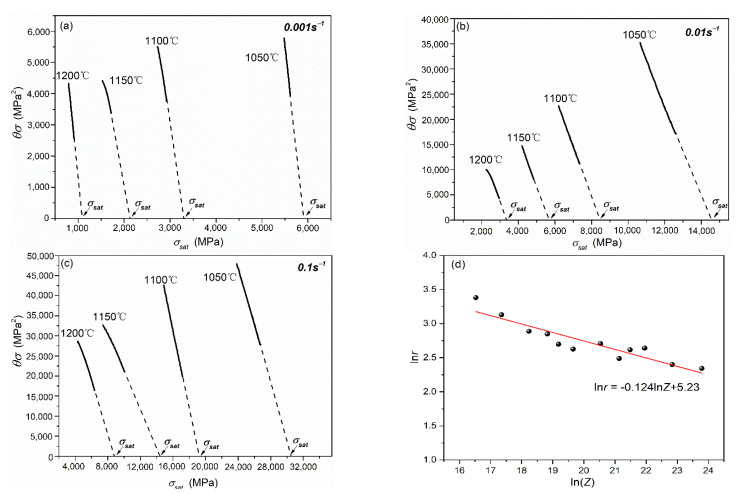
The plot of *σ*^2^ vs. *θσ* of the X70 pipeline steel at different deformation conditions: 0.001 s^−1^ (**a**); 0.01 s^−1^ (**b**); 0.1 s^−1^ (**c**); and the relationship between parameter *r* and *Z* parameter in the X70 pipeline steel (**d**).

**Figure 7 materials-15-07356-f007:**
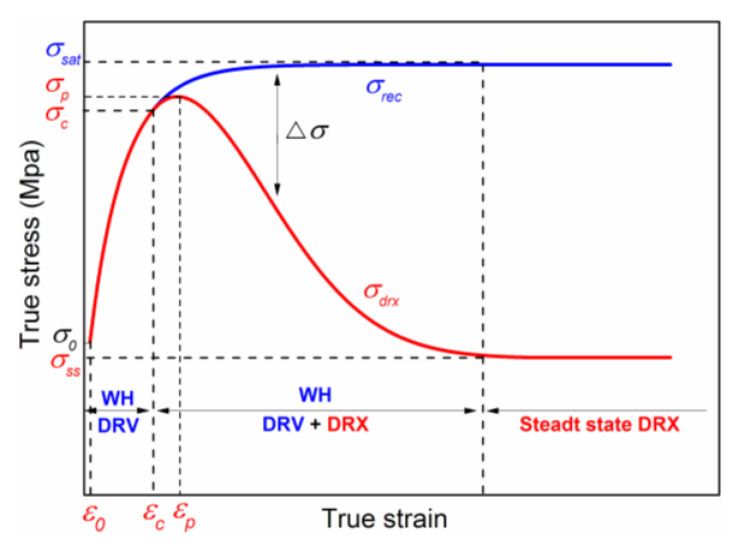
Schematic diagram illustrates the work-hardening curve *σ*_rec_ (the red line) from yield stress *σ*_0_ to saturation stress *σ*_sat_ in the entire strain domain, and a typical experimental DRX flow stress curve (the blue line) from yield stress *σ*_0_ to steady state stress *σ*_ss_.

**Figure 8 materials-15-07356-f008:**
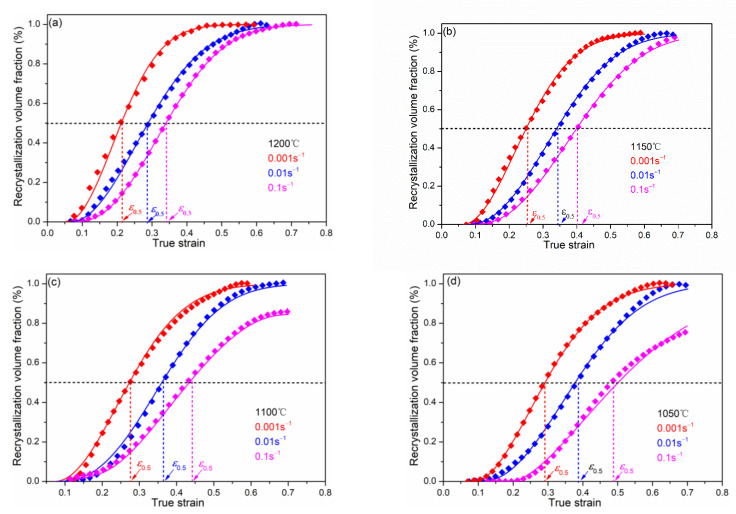
Comparison between the experimentally (symbols) and predicted (solid curves) volume fraction of DRX: (**a**) 1200 °C, (**b**) 1150 °C, (**c**) 1100 °C, (**d**) 1050 °C.

**Figure 9 materials-15-07356-f009:**
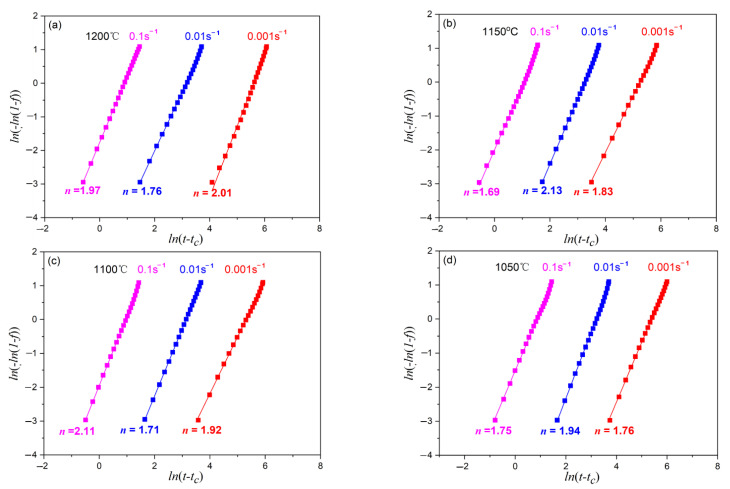
Value of Avrami index (*n*) at different deformation conditions. (**a**) 1200 °C; (**b**) 1150 °C; (**c**) 1100 °C; (**d**) 1050 °C.

**Figure 10 materials-15-07356-f010:**
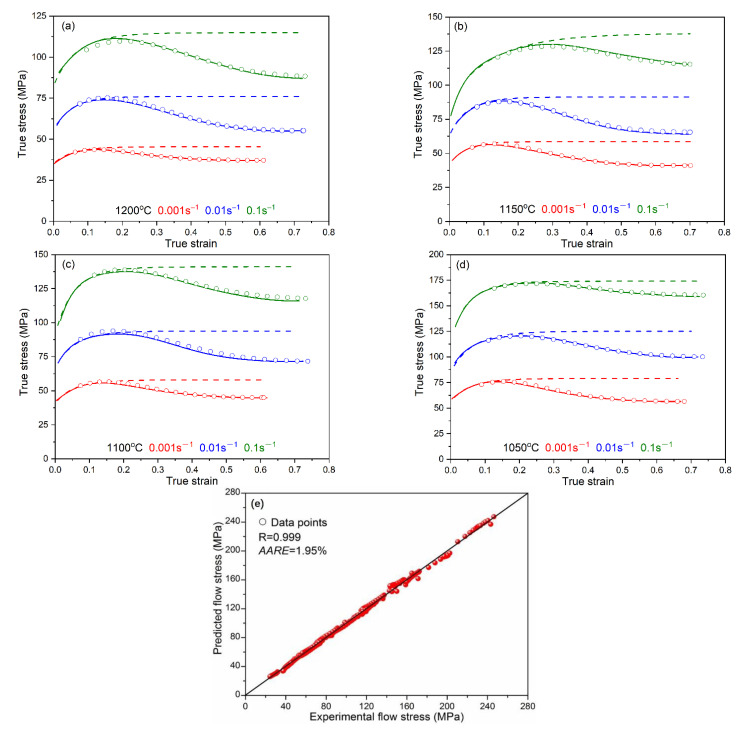
Comparison between the predicted and experimental flow stress–strain curves from constitutive equation at strain rate (**a**) 1200 °C, (**b**) 1150 °C, (**c**) 1100 °C, (**d**) 1050 °C, (**e**) correlation between the experimental and predicted flow stress data.

**Figure 11 materials-15-07356-f011:**
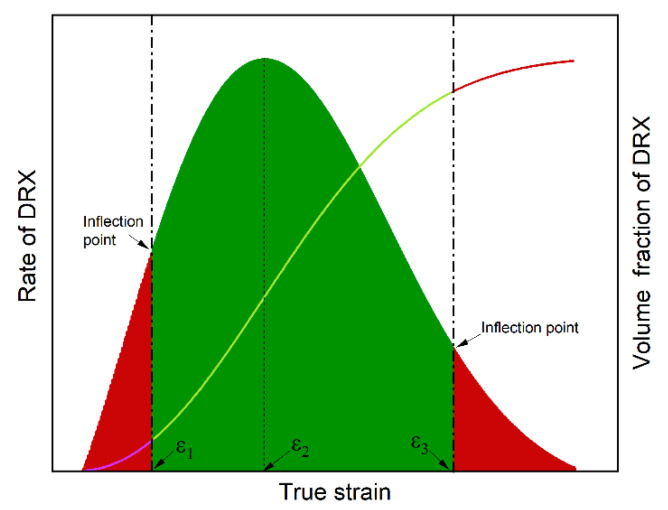
Schematic of velocity curve and volume fraction curve of DRX.

**Table 1 materials-15-07356-t001:** The relationship between characteristic stresses/strains and *Z*.

$\sigma_{x}=aZ^{b}$	$\varepsilon_{x}=cZ^{d}$
*σ*_p_ = 0.993 × *Z*^0.218^	*ε*_p_ = 0.0085 × *Z*^0.118^
*σ*_c_ = 0.884 × *Z*^0.218^	*ε*_c_ = 0.00434 × *Z*^0.118^
*σ*_ss_ = 0.775 × *Z*^0.218^	*ε*_ss_ = 0.6091 × *Z*^0.118^
*σ*_sat_ = 1.162 × *Z*^0.218^	*ε*_m_ = 0.0214 × *Z*^0.118^
*σ*_0_ = 0.606 × *Z*^0.218^	*ε*_0.5_ = 0.181 × *Z*^0.118^

## Data Availability

Some or all data, models, or code generated or used during the study are available from the corresponding author by request.

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
