# Peer review of "The Flow Stress–Strain and Dynamic Recrystallization Kinetics Behavior of High-Grade Pipeline Steels"

_materials, 2022, doi:10.3390/ma15207356_

Round 1
Reviewer 1 Report
The work presents the analysis of dynamic recrystallization kinetics utilizing mathematical models in X70 pipeline steel to optimize the design in the manufacture of pipeline steel.
The issue has been addressed in different types of steel to optimize manufacturing. It would have been interesting to analyze microstructure results with the proposed model.
Metals 2021, 11, 138. https://doi.org/10.3390/met11010138
J. Cent. South Univ. (2016) 23: 1007−1014 DOI: 10.1007/s11771-016-3149-2
The paper is an easy read, but the mathematical models used could be described in more detail. The presentation of the results could be improved by incorporating more discussion of the results with the various investigations in the area.
The conclusions agree with the results obtained, obtaining a kinetic model of dynamic recrystallization of the x70 pipeline steel.
Comments minor:
You must update the references; only reference 2 is less than five years old.
Describe in more detail the mathematical models used
Figure 2. Place the axes of the 4 graphs on the same scale.
Reference 5 is missing an l to the word steel
Author Response
Materials
Manuscript Number: materials-1863031
Full title: The flow stress-strain and dynamic recrystallization kinetics behavior of high-grade pipeline steels
Dear reviewer:
We are grateful to the reviewer for his/her positive opinion and for the valuable comments.
We have revised the manuscript in accordance with remarks made by the reviewers (see what follows). And the amendments and corrections to the original submission are marked up using the “Track Changes” in the revised manuscript.
Responses to the comments (concerns of the reviewers are indicated in italic).
1. You must update the references; only reference 2 is less than five years old.
Reply: We agree with the reviewer. The references have been updated in the manuscript.
2. Describe in more detail the mathematical models used
Reply: Following the reviewer’s suggestion, the mathematical models have been more detail described.
3. Figure 2. Place the axes of the 4 graphs on the same scale.
Reply: Thank you for pointing out this issue. The reason of the inconsistency of the ordinate in Fig. 2 is that since the Fig. 2b is a local magnification of the Fig. 2a, if the ordinate is consistent with Fig. 2a, the details of the curve in Fig. 2b could not be captured, resulting into the critical stress/strain point can not be accurately obtained. The ordinate of Fig. 2c is the product of the work-hardening rate and the flow stress, and differs greatly from Fig.2a and Fig. 2b. So, them can not be placed in one ordinate.
4. Reference 5 is missing an l to the word steel
Reply: The letter “l” omitted from reference 5 has been supplemented in this reviewed manuscript.
Reviewer 2 Report
This work pays attention to The flow stress-strain and dynamic recrystallization kinetics behavior of high-grade pipeline steels. Some proper conclusions have been obtained. However, a major revision is needed.
1. The language of this paper is good, and can be further improved.
2. A concise and factual abstract is required. It should contain the objective, methods, results, and conclusions, with emphasis on the results and conclusions.
3. Most of the provided references are old, it is better to use new and relevant references like the ones below:
- "High-temperature tensile characteristics and constitutive models of ultrahigh strength steel." Materials Science and Engineering: A 803 (2021): 140491.
- A comparative study of phenomenological, physically-based and artificial neural network models to predict the Hot flow behavior of API 5CT-L80 steel, Materials Today Communications 25, 101528.
- "Modeling Dynamic Recrystallization Behavior in a Novel HIPed P/M Superalloy during High-Temperature Deformation." Materials 15, no. 11 (2022): 4030.
- "Hot deformation behavior and related microstructure evolution in Au− Sn eutectic multilayers." Transactions of Nonferrous Metals Society of China 31, no. 6 (2021): 1700-1716.
4. What are the test procedures?
5. Considering that the discussion is about recrystallization, it is necessary to interpret the results with appropriate microstructure images of optical microscope and SEM (for example “the given kinetic model of DRX for guaranteeing uniform and fine grains.” “uniform and fine grains” requires appropriate microstructure interpretations)
6. Strain rate effects are very important in hot deformation, but these effects are not considered in this manuscript.
Author Response
Materials
Manuscript Number: materials-1863031
Full title: The flow stress-strain and dynamic recrystallization kinetics behavior of high-grade pipeline steels
Dear reviewer:
We are grateful to the reviewer for his/her positive opinion and for the valuable comments.
We have revised the manuscript in accordance with remarks made by the reviewers (see what follows). And the amendments and corrections to the original submission are marked up using the “Track Changes” in the revised manuscript.
Responses to the comments (concerns of the reviewers are indicated in italic).
1. The language of this paper is good, and can be further improved.
Reply: The language of the revised manuscript has been checked and further improved.
2. A concise and factual abstract is required. It should contain the objective, methods, results, and conclusions, with emphasis on the results and conclusions.
Reply: Thank you for pointing out this issue. The abstract of the revised manuscript has revised, in particular the section on results and conclusions.
3. Most of the provided references are old, it is better to use new and relevant references like the ones below:
Reply: Following the reviewer’s suggestion, the reference throughout in the revised manuscript has been updated, and most of them are within 5 years.
4. What are the test procedures?
Reply: Thank you for pointing out this issue. The experimental test procedures have been described in detail and a schematic of the test is also added in the experimental section of the revised manuscript.
5. Considering that the discussion is about recrystallization, it is necessary to interpret the results with appropriate microstructure images of optical microscope and SEM (for example “the given kinetic model of DRX for guaranteeing uniform and fine grains.” “uniform and fine grains” requires appropriate microstructure interpretations)
Reply: Thanks a lot for providing us such a valuable suggestion. Indeed, the occurrence of DRX often has the effect of refining the grain and the evolution process of DRX are generally studied by microstructure analysis. However, for high grade pipeline steel, when temperature is above the Ar1, the microstructure is a single-phase austenite. With the decrease of temperature (<Ar1), the austenitic phase occurs a ferrite/bainite phase transition, and the ferrite/bainite can be corroded by nitrate alcohol. However, the original austenitic grain boundaries are difficult to be clearly corroded, so this is why it is difficult to study the DRX of high-grade pipeline steel by using microstructure analysis. Thus, microstructure images of optical microscope and SEM are not given in the revised manuscript.
6. Strain rate effects are very important in hot deformation, but these effects are not considered in this manuscript.
Reply: Thanks a lot for providing us such a valuable reference. Indeed, the strain rate has a significant influence on the behavior of DRX. Especially under high strain rate conditions(>1s-1), High strain rate often causes the recrystallization mechanism to change, such as form discontinuous DRX (DDRX) to continuous DRX (CDRX). However, the strain rate in this paper is less than 1s-1, and under this condition, the recrystallization mechanism of high-grade pipeline steel is discontinuous DRX. On the other hand, this paper focuses on the prediction of flow stress and the volume fraction of DRX. The effect of strain rate on the DRX behavior of high-grade pipeline steel, especially under high strain rate conditions (>1s-1), will be discussed in detail in the next paper.
Round 2
Reviewer 2 Report
the manuscrpt can be accepted in present form